# Crop diversity induces trade-offs in microbial biopesticide susceptibility that could delay pest resistance evolution

Rosie M. Mangan [1]*, Matthew C. Tinsley[1], Ester Ferrari[1], Ricardo A. Polanczyk[2], Luc F. Bussière[3]

1 Biological and Environmental Sciences, School of Natural Sciences, University of Stirling, Stirling, United Kingdom, 2 Júlio de Mesquita Filho State University of São Paulo, Faculty of Agrarian and Veterinary Sciences of Jaboticabal, Jaboticabal, São Paulo, Brazil, 3 Biological and Environmental Sciences & Gothenburg Global Biodiversity Centre, The University of Gothenburg, Gothenburg, Sweden

* rosie.mangan1@stir.ac.uk

## Abstract

Pathogens often exert strong selection on host populations, yet considerable genetic variation for infection defence persists. Environmental heterogeneity may cause fitness trade-offs that prevent fixation of host alleles affecting survival when exposed to pathogens in wild populations. Pathogens are extensively used in biocontrol for crop protection. However, the risks of pest resistance evolution to biocontrol are frequently underappreciated: the key drivers of fitness trade-offs for pathogen resistance remain unclear, both in natural and managed populations. We investigate whether pathogen identity or host diet has a stronger effect on allelic fitness by quantifying genetic variation and covariation for survival in an insect pest across distinct combinations of fungal pathogen infection and plant diet. We demonstrate substantial heritability, indicating considerable risks of biopesticide resistance evolution. Contrary to conventional thinking in host-pathogen biology, we found no strong genetic trade-offs for surviving exposure to two different fungal pathogen species. However, changes in plant diet dramatically altered selection, revealing diet-mediated genetic trade-offs affecting pest survival. Our data suggest that trade-offs in traits not strictly related to infection responses could nevertheless maintain genetic variation in natural and agricultural landscapes.

## Author summary

Why don't all organisms in a population have the best genes to defend against infection? One potential explanation is that the ideal genotype needed to survive infection depends on the identity of other organisms (such as pathogens, predators or food) that a host interacts with in its environment. For example, host-pathogen evolutionary theory frequently assumes pathogen-driven

**Data availability statement:** Data and code underpinning this study are available on a freely accessible online repository. https://datastorre.stir.ac.uk/handle/11667/243

**Funding:** All authors were supported by a joint Newton Fund international partnership between the Biotechnology and Biological Sciences Research Council (BBSRC) in the UK and the São Paulo Research Foundation (FAPESP) in Brazil under BBSRC awards reference BB/R022674/1 & BB/S018956/1 and Grant 2018/21089-3, São Paulo Research Foundation (FAPESP). Additionally, LFB was supported by grants from Vetenskapsrådet (Sweden): 2021-05466, and the Carl Trygger Foundation (20:63). The funders had no role in study design, data collection and analysis, decision to publish, or preparation of the manuscript. Ester Ferrari and Rosie Mangan received a salary from the BBSRC award.

trade-offs such that genes enhancing defence against one pathogen may make hosts more susceptible to other infections. We investigated genetic variation for survival in a moth agricultural pest fed on different crop diets while exposed to fungal pathogen species that are used as biocontrol agents to protect crops. Moths best able to survive one fungal species tended also to survive quite well when exposed to a second. However, there was a trade-off between diets: moth genotypes that defended well against infection whilst eating one plant diet, tended to be more susceptible on a different plant diet. Our work addresses not only fundamental theory, but also the major practical challenge that global agriculture faces to control pests without driving resistance evolution. We conclude that farmers could manage resistance evolution through crop diversification that makes selection for resistance inconsistent.

## Introduction

Host populations in natural systems commonly harbour considerable genetic variation for susceptibility to pathogens [1–4]. Although classic evolutionary mechanisms like negative frequency-dependent selection (e.g., Red Queen dynamics) can explain some of this variation [5], genetic variation for pathogen susceptibility must also be maintained by other evolutionary forces, such as genotype by environment interactions (GEIs) [3,6]. When GEIs exist, a given genotype's effectiveness in conferring infection defence is conditional on specific environmental contexts [7]. This raises a crucial question: which aspects of environmental heterogeneity play the strongest role in driving inconsistent selection on gene loci influencing pathogen susceptibility [3,8]?

Human activities, particularly in agriculture, often reduce environmental heterogeneity. For example, extensive monocultures diminish landscape variation and increase pest outbreaks [9]; alongside chemical pesticide use, this selects for resistant pests and increases crop damage [10,11]. Consequently, there is growing interest in more ecologically sustainable insect pest control products, such as microbial biopesticides containing living organisms [12–14]. As the use of these microbial biopesticides increases [15], so too will selection pressures to evolve resistance against them [16].

Here, we consider two concepts related to resistance. In immunology, resistance describes the ability to prevent infection or suppress growth of pathogens [17]. In agriculture, resistance evolution refers to genetic changes in pest populations that impair efficacy of a pest control product [18]. These definitions overlap in the study of infection by pathogens used in biocontrol. In both cases, resistance is a quantitative trait, such that the degree of defence may be determined both by the effects of different alleles within individuals and by the frequency of those alleles at a population level [18].

Traditionally, strategies to manage resistance to synthetic pesticides and genetically modified crops involve weakening selection for resistance, for example by minimising pesticide application, employing crop refuges, or alternating the use of products with different modes of action [19,20]. An additional approach is Negatively Correlated Cross-Resistance (NCCR), which aims to exploit the fact that resistance to

one pesticide product sometimes trades-off with resistance to another [21]. NCCR has been used successfully in the management of insecticide resistance for onchocerciasis vector control [22]. However, despite its promise, NCCR has not been widely used in agriculture [8,23] for at least two reasons. First, the negative fitness correlations between pesticides needed for NCCR rarely exist. Second, trade-offs for performance across pesticides are not immutable, and resistance trade-offs can themselves evolve [24,25], presumably because resistance to synthetic pesticides often depends on a small number of independent loci [26], and recombination could rapidly generate genotypes that are resistant to multiple products.

Here we re-explore the principles behind NCCR in the context of biopesticides formulated from living microbes; this is especially timely during the ongoing transition to more environmentally sustainable pest control in global agriculture [12]. In contrast to synthetic pesticides, we believe there is much greater potential for NCCR involving biopesticides formulated from living microbes. On account of the greater biomolecular complexity involving interactions with living organisms rather than individual chemical compounds, defence against microbes is expected to be more genetically complex than for synthetic insecticides or genetically modified crops [8,26–29].. Whilst resistance to microbes used in biocontrol sometimes involves a small number of gene loci, in most cases where this has been studied in detail, the genetic basis has turned out to be relatively complex [30–34]. Such complex genetic architecture should make it more difficult for recombination to resolve the trade-offs required for NCCR [8].

Resistance trade-offs may arise due to specific genetic interactions between hosts and pathogens: genotypes defending against one pathogen species or strain can increase susceptibility to another [35]. Leveraging such GEIs could help manage resistance if biopesticides containing different microbial pathogens are used in rotation. In addition to the pathogens themselves, multiple environmental factors, especially variable temperatures, are known to drive GEIs related to pathogen susceptibility [3,6,36]. Farmers cannot control temperatures to mitigate resistance risks, yet they do control crop selection, which for polyphagous pests dictates the pests' diet and can substantially influence infection defence efficacy and immune function [37]. For example, the efficacy and costs of resistance to Bt are sensitive to both the diet that the pest feeds on [38,39] and co-exposure to other antagonists such as additional microbes and parasites [40,41]. However, environmental differences do not always weaken genetic correlations [36] and whether diet can impose heterogeneous selection on genotypes promoting survival through GEIs remains underexplored. We set out to test the extent to which heterogeneity in both the pathogens used in biopesticides and the crops grown in agricultural landscapes might be used to manage the threats of resistance to microbial biopesticides evolving.

In this study, we focus on the noctuid moth, *Helicoverpa armigera* (cotton bollworm), a major global agricultural pest with a history of developing resistance to multiple control tactics [42–45]. We study fungal biological control agents because they are especially promising in the context of GEIs due to the complexity of their infection process: penetrating the insect cuticle, replicating inside the host, and ultimately killing the host to facilitate onward transmission [46]. The genomes of entomopathogenic fungi encode multiple virulence mechanisms to infect hosts, manipulate their physiology and subvert their defences [47], while host survival to these pathogens is typically highly polygenic [29]. Previous studies of the genetic control of fungal pathogen infection susceptibility also show that allelic variation has a graded rather than binary effect on survival probability in natural populations of insects [4,28,29].

This study has twin aims, one fundamental and one applied. From a fundamental perspective, we asked: to what extent can the pervasive presence of genetic variation for infection defence be explained because GEIs drive inconsistent selection under different environmental conditions? Which is the more powerful driver of inconsistent selection: variation in the identity of the pathogen, or variation in the diet the animal consumes during infection? We also exploited this evolutionary science for an applied goal. We aimed to quantify the risks that the crop pest *H. armigera* can evolve resistance against fungal biopesticides by quantifying the standing genetic variation for survival following pathogen exposure on which selection could act. Furthermore, we sought to determine how GEIs might be exploited by farmers to make selection for surviving biopesticide exposure inconsistent, thereby managing the threat that pest evolution will render these ecologically sustainable pest control products ineffective.

## Results

### Survival consequences of changes in plant diet and pathogen treatment

Our dataset quantified survival in 3811 *H. armigera* larvae. The two pathogen exposure treatments (*Beauveria bassiana* or *Metarhizium anisopliae*) induced markedly higher larval mortality than in uninfected larvae regardless of the leaf diet (Fig 1). Also, background mortality varied for larvae feeding on the three different food plants: larvae survived best when fed soybean, whereas those reared on tomato or maize were more likely to die (Fig 1, S1 Table).

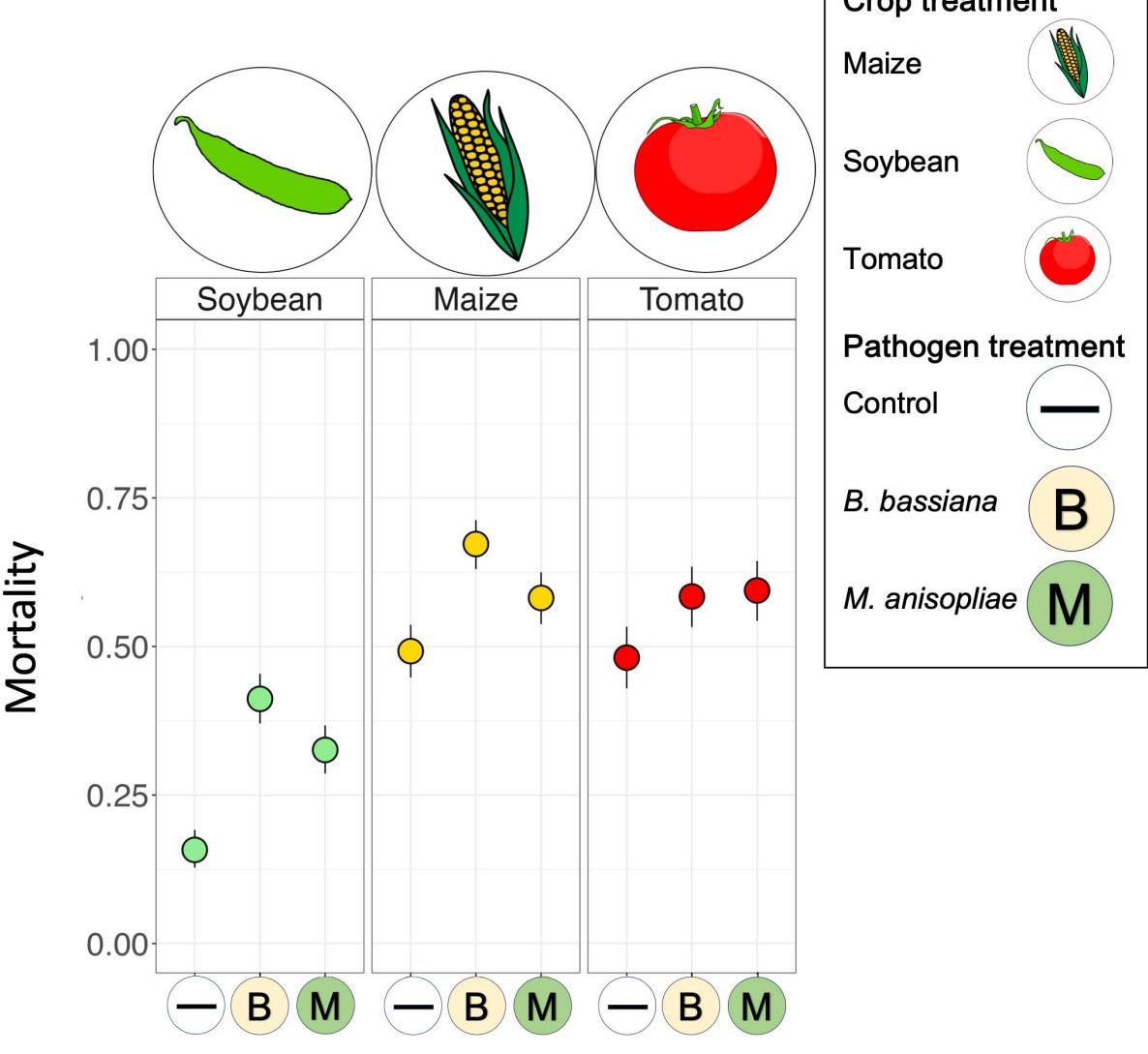

**Fig 1. Relative ability of fungal isolates to kill *H. armigera* larvae depended on crop leaf diet.** Dots indicate mean mortality at day 14 post-infection; whiskers give 95% binomial confidence limits for each combination of pathogen treatment (on the x-axis) and plant leaf diet (in panels). Total n = 3811 larvae across the 9 treatment combinations.

We conducted our experiment using two experimental blocks; there was a difference in overall mortality between the blocks but the patterns among treatments were remarkably consistent (S1 Fig). Interestingly, the ability of fungi to kill larvae was driven by the combination of fungal isolate and larval crop diet (plant:pathogen treatment interaction, parametric bootstrap p-value = 0.003; S1 Table): whilst *B. bassiana* caused greater mortality than *M. anisopliae* in larvae feeding on soybean and maize leaves, this virulence advantage disappeared for larvae on tomato (Fig 1).

## Genetic variation for larval survival ability

Our experiment used a half-sib breeding design to quantify additive genetic variation for ability to survive pathogen exposure; the dataset included offspring from 37 *H. armigera* sires, collectively mated to 58 dams (mean number of dams per sire: 1.57; 1 dam: 18 sires; 2 dams: 17 sires; 3 dams: 2 sires). Risks of pest resistance evolution in response to fungal biopesticide application may be greatest if pest populations harbour pre-existing additive genetic variation for infection susceptibility on which selection can act. The half-sib *H. armigera* families in our experimental design varied greatly in ability to survive fungal pathogen exposure (Fig 2).

To work out what fraction of this between-family phenotypic variation was due to additive genetic effects, we calculated heritabilities in each pathogen-diet treatment. Our Bayesian analyses provide posterior distributions of all model parameters (as well as measurements like heritability that can be computed from posterior samples); the dense regions of these distributions have more statistical support (see Methods for more details). This analysis indicates substantial heritability in all treatments; the median posterior heritability ranged from 0.46 - 0.77 depending on the treatment (Fig 3).

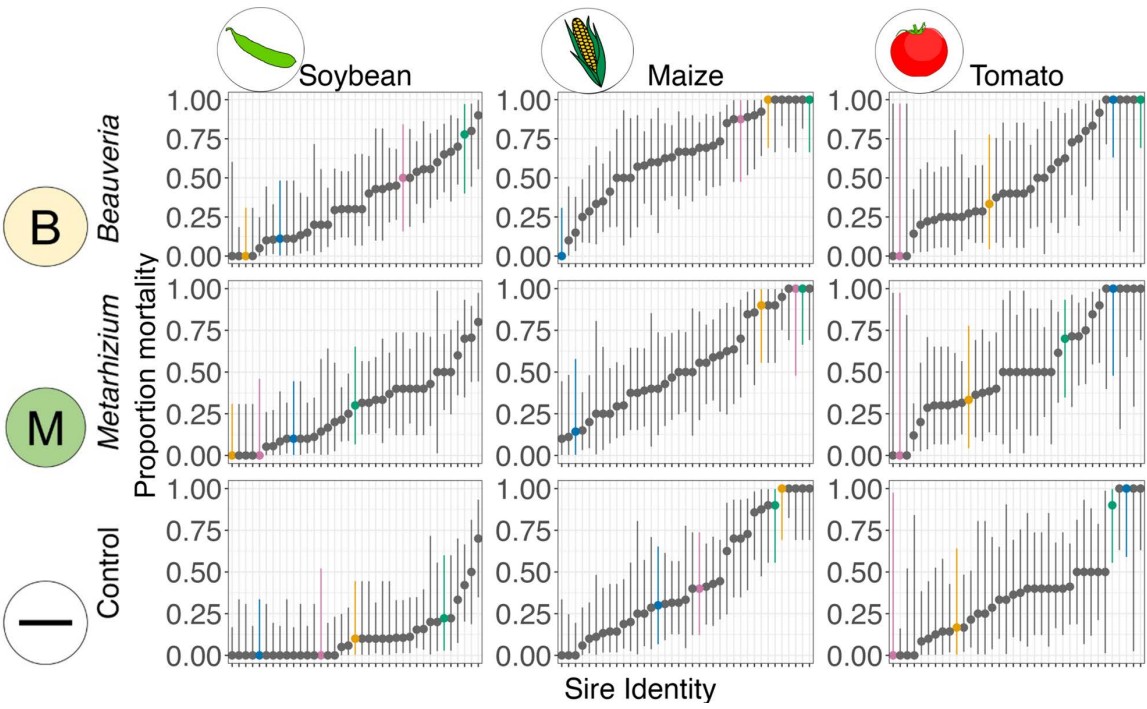

**Fig 2. Diet and infection treatment strongly alter the relative fitness of different half-sibling families.** Each point and whisker represents mean mortality and 95% binomial CI for sire half-sibling families 14 days post-exposure, depending on plant diet (columns) and pathogen treatment (rows). Four of these families are highlighted consistently across panels (see coloured points) to illustrate contrasting patterns of performance across habitats. The sires are arranged along the x-axis in order according to their rank performance in each panel. The phenotypic patterns in this plot are formally decomposed into the additive genetic components in Table 1 and Fig 4.

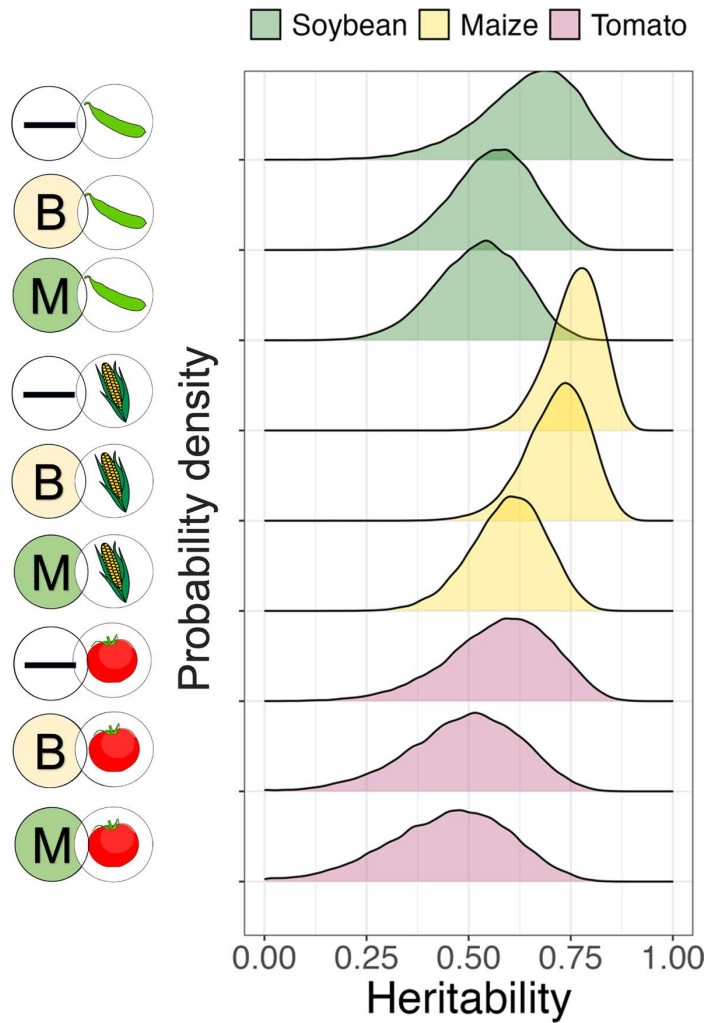

**Fig 3. A substantial fraction of variation in larval survival in each treatment is heritable.** Posterior distributions from a Bayesian analysis of heritabilities (the proportion of variation in survival that is additive and genetic) in each of the nine combinations of plant diet and infection treatment. The density of each distribution indicates the weight of posterior evidence for a particular heritability value (on the x-axis). These estimates were obtained from 58 half-sib families of *H. armigera* fed on one of three leaf diets (soybean, maize, or tomato) and exposed to one of three pathogen-exposure treatments (M = *M. anisopliae*, B = *B. bassiana*, — = Control).

Counterintuitively, the heritabilities for survival in the fungal exposed treatments were not noticeably higher than those in the control treatment (for any plant diet), which suggests that the standing genetic variation we observed is not solely related to pathogen susceptibility. Instead, some of the genetic variation seems also to relate to general performance in our experimental conditions (e.g., feeding on leaves of the three crops provided) even in the absence of pathogens.

### Genotype by environment interactions for infection-susceptibility

The high heritabilities we observed were driven by extreme survival differences between families: by 14 days post infection, in most pathogen treatments, families varied between some with close to 0% mortality and some with close to 100% mortality (Fig 2). Those families that survived best under one combination of plant diet and infection treatment often performed relatively poorly in other treatment combinations (S2, S3, and S4 Figs). Indeed, the plant diet and pathogen treatments

sharply affected the relative susceptibilities of families to infection, as expected if there are strong genotype-by-environment interactions affecting allelic fitness (see the rank performance of highlighted families in Fig 2). We further illustrate the change in family performance by ordering paternal families according to their mean survival whilst feeding on soybean and exposed to *B. bassiana*, plotted in the upper left panel of S5 Fig. These visual inspections suggest that genetic correlations across treatments were lower (indicating potential fitness trade-offs) when treatment contrasts involved changes in the diet (S3, and S4 Figs) than when they involved changes in pathogen treatment (S2 Fig).

## Genetic correlations between larval survival ability in different crop-pathogen treatments

To formally compare how crop and biopesticide changes affect the magnitude and direction of selection for biopesticide resistance, we calculated genetic correlations, which quantify the degree to which alleles for survival in one environment will also be favoured in a second. A perfect genetic correlation ($r_g = 1$) means that environmental changes do not alter relative allelic fitness. Correlations between zero and one imply GEIs will delay responses to selection if environments change, and correlations below zero indicate that the direction of selection is reversed across treatments [48].

The magnitude and sign of cross-environment genetic correlations for survival depended markedly on whether the treatments differed in crop diet or pathogen exposure. Table 1 contains a summary of the single-model G-matrix (the matrix of median variances and covariances) on the upper half-diagonal, and the median genetic correlations on the lower half-diagonal. However, as for heritability estimates, the total evidence under the posterior is better illustrated by the ridge-plots in Fig 4.

Surprisingly, changing pathogen treatment without changing the host plant (either changing between fungal genera or from control to a pathogen infection treatment, Fig 4, red shaded density ridges) depressed genetic correlations only modestly below 1 (mean = 0.48, bootstrapped 89% HDI = 0.39 - 0.58). By contrast, changing the plant diet depressed genetic correlations far more, with many such correlations estimated below zero, revealing crop-mediated genetic trade-offs for infection susceptibility (Fig 4, green ridges, mean = -0.10, 89% HDI = -0.21 – 0.01). Counterintuitively, simultaneous change of both pathogen and diet treatments (right-most panel in Fig 4) provided no obvious further depression in the genetic correlation (mean = -0.09, 89% HDI = -0.17 – 0.01): when the genetic correlation involved environments that differed in both dimensions, the genetic correlations were barely distinguishable from those involving only plant diet changes.

**Table 1. Changes in plant diet are associated with strong genetic trade-offs.** Below are summaries of genetic variances (orange, diagonal), covariances (blue, upper off-diagonal), and correlations (green, lower off-diagonal) for mortality across 9 combinations of plant and pathogen exposure treatments. Within-plant diet correlations are shaded in darker green to call attention to their consistently higher values than the cross- plant diet correlations.

| | | Soybean | | | Maize | | | Tomato | | |
|---|---|---|---|---|---|---|---|---|---|---|
| | | Control | *Beauveria* | *Metarhizium* | Control | *Beauveria* | *Metarhizium* | Control | *Beauveria* | *Metarhizium* |
| Soybean | Control | 2.006 | 0.399 | 0.532 | 0.049 | -0.15 | -0.356 | -0.385 | 0.056 | -0.113 |
| | *Beauveria* | 0.252 | 1.276 | 0.69 | -0.561 | -0.251 | -0.328 | -0.447 | -0.304 | -0.242 |
| | *Metarhizium* | 0.369 | 0.588 | 1.113 | -0.617 | -0.387 | -0.586 | -0.429 | -0.178 | -0.181 |
| Maize | Control | 0.01 | -0.299 | -0.349 | 3.152 | 1.781 | 1.473 | 0.311 | 0.645 | 0.427 |
| | *Beauveria* | -0.082 | -0.151 | -0.243 | 0.637 | 2.569 | 1.318 | 0.056 | 0.151 | 0.078 |
| | *Metarhizium* | -0.225 | -0.264 | -0.481 | 0.697 | 0.688 | 1.472 | 0.219 | 0.225 | 0.224 |
| Tomato | Control | -0.253 | -0.359 | -0.369 | 0.149 | 0.029 | 0.158 | 1.457 | 0.321 | 0.342 |
| | *Beauveria* | 0.028 | -0.302 | -0.198 | 0.375 | 0.098 | 0.196 | 0.268 | 1.01 | 0.496 |
| | *Metarhizium* | -0.104 | -0.266 | -0.219 | 0.262 | 0.056 | 0.209 | 0.305 | 0.534 | 0.877 |

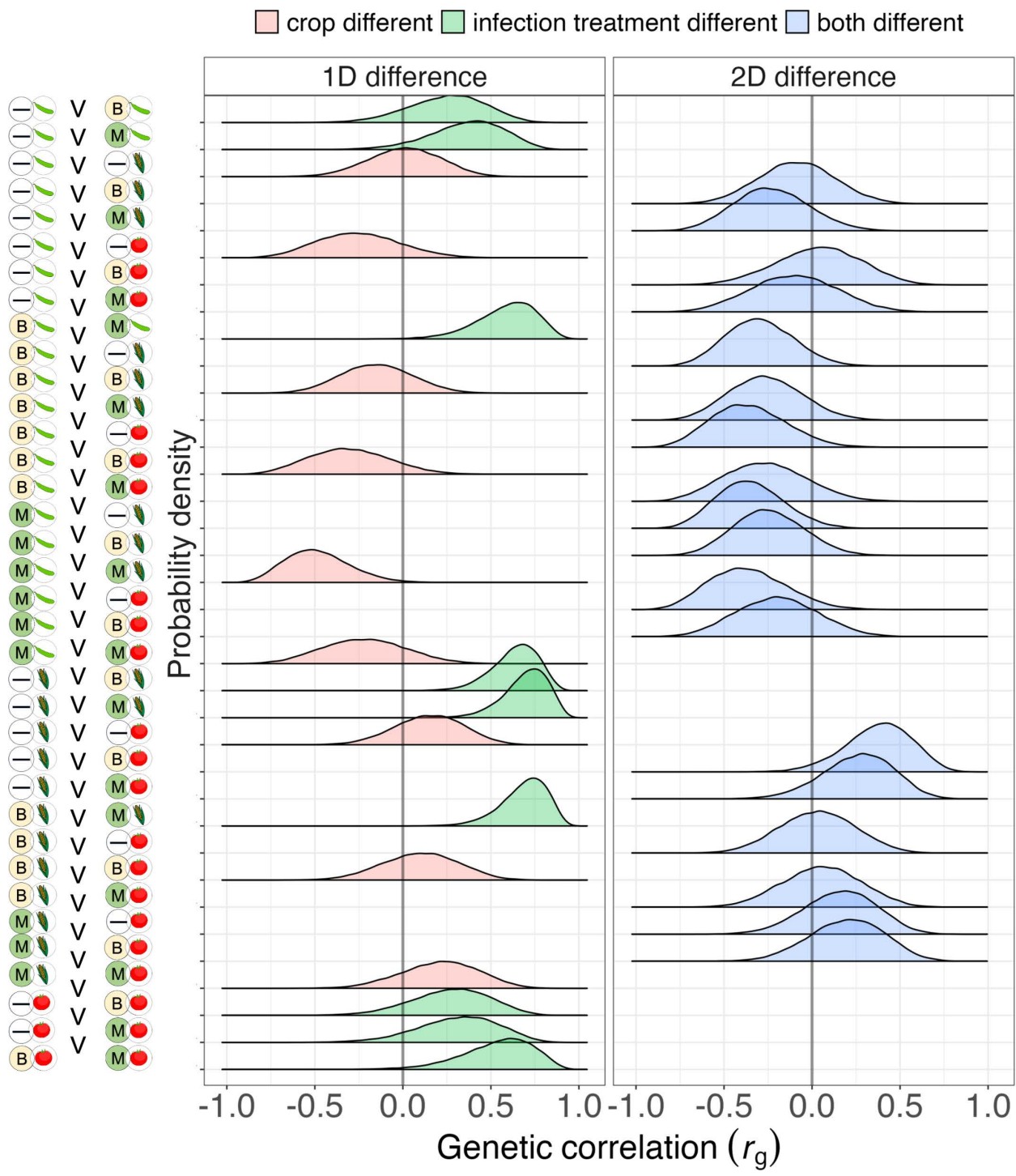

**Fig 4. Genetic correlations are lower when measured across environments that differ in plant diet.** Posterior distributions for cross-environment genetic correlations for mortality in *H. armigera* larvae grown in 9 different combinations of plant diet (soybean, maize, and tomato) and pathogen treatment (control, *Beauveria,* and *Metarhizium*). The 36 posteriors are clustered in the figure above depending on the axes of environmental difference, with environments differing in only 1 dimension on the left, and those differing in 2 dimensions on the right. Summaries of central tendency in the genetic correlations appear in the lower off-diagonal of Table 1.

## Discussion

Pathogen exposure can exert strong selection on host populations. However, the extent to which this selection is shaped by external factors, such as host diet and pathogen identity, is not generally clear. Motivated by the need to assess how emerging risks of biopesticide resistance evolution could be managed, our study aimed to assess whether diet and pathogen differences can drive GEIs for survival. We found that genotypes of *H. armigera* vary substantially in their ability to survive infection by fungi that are used as biopesticides, as demonstrated by considerable heritabilities in all combinations of crop and pathogen treatment. This high level of standing genetic variation presents a clear risk of resistance evolution against fungal biopesticides used in agriculture. We also reveal that altering the crop that larvae feed on can generate strongly inconsistent selection (evidenced by frequently negative genetic correlations for larval survival across plant diets), whilst changes in the identity of the fungal pathogen (where we found moderately positive genetic correlations) alter selection to a lesser extent.

The crop plant on which *H. armigera* larvae were feeding had a marked effect on survivorship. Larval survival in the uninfected control treatment was high on soybean leaves; however, approximately 50% of larvae died during two weeks on both the maize and tomato leaf diets. *H. armigera* feeds on over 100 plant species but not all are ideal for its survival; indeed secondary phytochemicals, nutrient deficiencies, and physical leaf defences can impair fitness [49,50]. Some polyphagous herbivorous insect species, like *H. armigera*, are composed of many different genotypes that specialise on different plant species [51], leading to genetically distinct populations based on crop type [52]. The population of *H. armigera* we studied may be better adapted to feed on soybean than the other two diets. We observed significant heritability for the ability to survive on all three plant diets under control conditions; this mirrors previous work studying *H. armigera* larval development on different chickpea varieties [53].

When we exposed larvae to *B. bassiana* or *M. anisopliae* fungal pathogens, mortality rates increased compared to the control treatment. However, the magnitude of this infection-induced mortality depended on the precise combination of pathogen and plant diet: *B. bassiana* caused higher mortality than *M. anisopliae* when larvae ate soybean and maize, whereas for larvae feeding on tomato leaves the two pathogens caused similar mortality. It is not clear why diet should modulate host infection susceptibility differentially depending on the identity of the pathogen; however, diet composition is well established to influence disease resistance [54]. Our data demonstrate that, for polyphagous pests, the efficacy of particular biopesticides may vary depending on the crop species a farmer grows, an observation that may have important consequences for the agricultural industry.

Pest insects frequently evolve resistance against chemical pesticides and genetically modified crops [20,26,55], yet the potential for pest evolution to diminish the efficiency of biopesticides formulated from living pathogens remains underappreciated, despite notable examples [8,56,57]. The risk of resistance will depend largely on the existence and magnitude of pre-existing genetic variation for survival in the presence of pathogens, but such measures are unavailable for most pests. We demonstrate substantial heritabilities (the fraction of phenotypic variation that is straightforwardly inherited) for pathogen susceptibility and that these estimates vary depending on the precise combination of pathogen treatment and plant diet. We note that the magnitude of heritability for survival in these pathogen exposure treatments is not solely driven by genetic variation for infection susceptibility, because we also observed substantial heritabilities in the absence of infection in the control treatments. Nevertheless, a significant proportion of the mortality in our infection treatments was specifically driven by pathogen exposure, supporting our argument that diet mediates the efficacy of genetic variants influencing pathogen susceptibility. Heritabilities in the wild, under heterogeneous field conditions will certainly be lower than in our standard laboratory conditions, because the impact of environmental variation on infection susceptibility will be more important in field conditions. In addition, we deliberately chose doses of fungal spores that caused intermediate survival to facilitate the detection of changes in variation across treatments. Nevertheless, the mean mortality rates we observed were representative of those commonly achieved by farmers when using biological control [58]. Higher doses that create greater mortalities may expose less genetic variation, both because some of the variation important at low doses becomes

irrelevant, and because of the relatively greater importance of binomial sampling at more extreme values of mortality [59]. Regardless, our results clearly support considerable standing genetic variation for survival in the presence of biological antagonists and justify further work to quantify the risk of evolution in response to biocontrol agents [60].

Classic concepts in host-pathogen evolution often assume that susceptibility to one pathogen genotype comes at the cost of impaired defence against others [61]. However, whether fitness traits generally trade-off against one another or are positively associated is extensively debated in life history research [62,63]. In the context of biopesticides, such trade-offs would mean that rotations of agricultural products containing different pathogens might be a highly effective resistance management approach [8,57]. We observed little evidence of susceptibility trade-offs driven by pathogen identity. Our analysis revealed universally positive genetic correlations for host fitness between infection treatments containing either *B. bassiana* or *M. anisopliae*, meaning that insect genotypes best able to defend against one pathogen were generally well-equipped to defend against the other. These pathogens are both fungi, which might be more likely to yield the observed positive correlations than comparisons between more phylogenetically distant pathogens. However, two studies in *D. melanogaster* found positive correlations between resistance to a fungal pathogen and a bacterial pathogen: *M. anisopliae* and *Pseudomonas aeruginosa* [29], *B. bassiana* and *Lysinibacillus fusiformis* [64]. The lack of defence-specificity in these cases may stem from the absence of a closely coevolved relationship between host and parasite, a situation likely common among microbial biopesticides used in crop pest control, as well as in the case of newly emerging infectious diseases.

Although there was little evidence of trade-offs between defence against *B. bassiana* and defence against *M. anisopliae*, the genetic correlations in susceptibility between these two pathogens were not very strong (i.e., far from a perfect correlation of 1). The rank order of genotype susceptibility was only modestly conserved between the pathogens, which is clear evidence for pathogen-mediated GEIs. Thus, even in the absence of strong trade-offs, evolutionary responses in populations exposed to different fungal pathogens in sequence will be less rapid than when selection is driven consistently by a single pathogen genotype.

Evolutionary ecology theory frequently assumes that effective parasite defence is costly, and therefore not favoured by selection when parasites are encountered infrequently. Indeed, compelling evidence for this assumption exists for some parasites [65]. However, the mechanistic basis of resistance varies greatly across host-parasite systems, and not all mechanisms of resistance need be generally costly. If strong broad costs for infection defence occurred in the *H. armigera* – fungus system, these might be evident as negative genetic correlations between the control and fungus-exposed treatments. Surprisingly, the genetic correlations for survival are roughly the same whether they are between treatments involving two different pathogens, compared to cases where the contrast is between a pathogen treatment and the control treatment. From an applied perspective, this predicts that farmers rotating between biopesticides containing different microorganisms would be just as evolutionarily sustainable as alternating periods of biological control with periods of no pest control at all. If this finding proves to be general, there may be no benefit to biopesticide-free refugia on a sufficiently diverse landscape treated with multiple biopesticides. However, it is worth remembering that we modelled a single and simplified response variable: the ability to survive 14 days after infection. One might rightly question the extent to which this response adequately reflects selection across the entirety of larval development, and we invite more work on the genetic architecture of multiple life history traits across multiple environments, even as we respect the appreciable samples and processing time that such experiments and analyses will require.

Host-parasite theory focuses on parasite identity and to a lesser extent environmental temperature as drivers of inconsistent selection that prevent genetic variation for infection defence being efficiently purged by selection [3]. However, our data demonstrate that variation in diet can generate previously underappreciated heterogeneous selection for pathogen resistance. In striking contrast to the modestly weakened genetic correlations across pathogen treatments, genetic correlations for post-infection survival were strongly depressed by changes in plant diet and were far more likely to produce negative genetic correlations. Indeed, shifting from soybean to another host plant consistently produced the most pronounced negative genetic correlations observed in our study. Genetic correlations for survival between maize

and tomato diets were also low, but not as low as those involving soybean and another crop. Whether this pattern is due to dietary specialisation for soybean in our moth population remains unclear. In the field, the specific nature of the habitats and pest populations will dictate whether trade-offs across heterogeneous patches can reverse biopesticide resistance evolution, and to what extent. In contrast to our findings, a study on aphids and their wasp parasitoids found little support that plant species altered the susceptibility of particular aphid genotypes to parasitism [66]. There are still too few studies of diet-induced GEIs to generalise, but the exciting possibility for inconsistent selection on parasite defence driven by diet variation suggested by our work and other research [41] invites further study.

What phenotypic differences account for observed survival variations among genotypes, and what biological process could explain why the fitness of genotypes to defend against infection is strongly dependent on the crop diet on which larvae feed? Microbial symbionts can have strong effects on the ability of insects to defend against infection [66] and to feed on particular plant diets [67,68]. However, our estimates of genetic (co)variances come from the male parental contribution to offspring phenotypic variation; as we expect that most gut, or other, microbial symbionts would be maternally inherited [69], we think that any non-genetic microbiome contributions to our estimates are probably small. Instead, a possible mechanism for diet-induced changes in pathogen resistance involves macronutrient-sensitive biochemical pathways. For example, dietary protein: carbohydrate ratios influence the ability of insects to upregulate immune responses and survive infection [54]. Whatever the mechanisms, and regardless of whether they involve pathways conventionally associated with immune function, the genetic differences we observed provide the prospect for crop-sensitive adaptive evolution in response to biopesticide exposure.

In demonstrating that crop heterogeneity can alter the intensity and direction of multivariate selection for survival in the presence of biopesticides, we have addressed a global food security question using theory from fundamental ecology and evolutionary science. Our research has important implications for agriculture and points to some questions requiring further research. We previously proposed that farmers could combat threats of pest evolution by engineering additional environmental heterogeneity into agricultural landscapes [8], especially through use of spatial matrices or temporal rotations of different biopesticides and crops. The motivation for our suggestion (managing resistance evolution) contrasts with those for other agricultural diversification practices, such as intercropping and push-pull strategies, which promote ecological benefits or suppress pest populations. In the present study, we show that some dimensions of landscape heterogeneity could change the intensity and direction of selection on pest survival but that not all dimensions are equally effective. For instance, our findings show modest negative genetic correlations across some plant diets, with an average of −0.10 for changes in plant diet alone, and a statistically indistinguishable −0.09 when diet differences are combined with different pathogen treatments. The limited strength of these negative genetic correlations indicates that reversing genetic adaptations through evolutionary processes across multiple traits may require several generations in varied habitats. This underscores the need for a broad spectrum of divergent selection strategies to prevent biopesticide resistance evolving, beyond the limited number of pest control products that were initially envisioned to induce negatively correlated cross resistance for chemical pesticides [21].

Our results bolster the theoretical prediction that divergent selection is a pivotal force in maintaining genetic variation for key life history traits. However, our data call into question the prevailing expectation among many evolutionary ecologists that trade-offs in host resistance to different pathogens have a prime role in maintaining genetic variation for pathogen defence traits. Instead, changes in diet can alter the fitness of pathogen resistance genotypes. This should prompt further studies of how other aspects of heterogeneous habitats shape temporal and spatial variation in selection.

## Methods

### Ethics Statement

All experimental protocols involving live insects were approved by the University of Stirling's ethical review board, adhering to UK standards for research.

## Plants

Soybean (*Glycine max* (variety Summer Shell)), tomato (*Solanum lycopersicum* (variety Roma)) and maize (*Zea mays* (variety Tramunt)) were grown from seed (Tamar Organics, UK) in a controlled environment facility at the University of Stirling (16:8 hr L:D photoperiod with compact-fluorescent lamps; 24°C/16°C during L/D; 70% R.H.). Seeds were placed individually in small pots (5 cm × 5 cm × 5 cm) containing approximately 150 g John Innes Seed Compost to germinate. Germinated seeds were then transferred to larger pots (12 cm × 15 cm) in approximately 700g John Innes No 2 compost.

## Preparation of fungal material

We used two fungal isolates from Campinas Biological Institute (Brazil) that are virulent against *H. armigera* (IBCB 1363 (*B. bassiana*) and IBCB 425 (*M. anisopliae*) [70]. Fungal material was grown on potato dextrose agar with chloramphenicol ($5 \times 10^{-5}$ g ml$^{-1}$). Agar plates were incubated for 10 days (25°C, 24 hr dark), then dried at room temperature for approximately 10 days; plates were rotated periodically to ensure even drying. Then, sporulating fungal material was scraped from the plates and spores dried further on silica gel in a fridge before being suspended in sunflower oil. These formulations were vortexed, and then briefly agitated with a probe sonicator to break up spore masses. Spore suspension concentrations were determined using a haemocytometer and adjusted to $2 \times 10^7$ conidia ml$^{-1}$.

## Experimental system

All insect culturing and experiments were conducted in the quarantine facility at the University of Stirling in controlled environment rooms. *Helicoverpa armigera* pupae were sourced from Andermatt Biocontrol AG, Switzerland. The insects used in this experiment originated from six separate consignments of pupae from Switzerland. Whilst the precise details of this source population are commercially sensitive, it is very large and used for the industrial-scale production of baculovirus biopesticides. We also know that this population exhibits substantial genetic variation from this study (see Results) and other experiments in our laboratories. On arrival, pupae were washed in 1% (w/v) copper sulphate solution, sexed and placed under conditions of reversed photoperiod (10 hrs dark between 03.00 and 13.00 hrs). Male pupae were held at 27°C, and females at 25°C to hasten the emergence of adult males and ensure sexual maturity synchrony of the sexes. Single mating pairs (one female <24hrs old and one male >3 days old) were placed in ventilated plastic containers (55 mm (l) × 55 mm (w) × 60 mm (h)) and provided with vitamin solution [71]. Males always originated from the shipment preceding the females to ensure outcrossing. All experimental protocols involving live insects were approved by the University of Stirling's ethical review board, adhering to UK standards for research.

## Experimental design

The experimental design (Fig 5) involved studying survival of half-sibling larvae in each of nine different experimental treatments (combining three diets, two pathogen infection treatments and an uninfected control). To examine genetic variation in defence traits, we mated each of 37 sires with up to three dams, resulting in 37 paternal half-sib families [72].

Mating pairs were observed for female receptivity to males: for females, calling (pheromone release) signified the attainment of reproductive maturity, a behaviour which was immediately identifiable and characterised by the female's extruded ovipositor. The maturity status of males was tested by examining their response to a calling female. Male mating behaviour consisted of brush extension and swiping movements of the abdomen directed at the calling female. Males that did not attempt to mate were classified as immature and tested again in 24 hrs. Unreceptive females moved away from an approaching male, withdrew the ovipositor, and flexed the abdomen resulting in the tip held beneath the female and inaccessible to male claspers. If a male was successful grasping an unreceptive female's abdomen, females initiated violent wing fanning to escape. This behaviour starkly contrasted with that of receptive females, who ceased all activity about 15 seconds after pairing with a male. Mating pairs were allotted 15 min to mate.

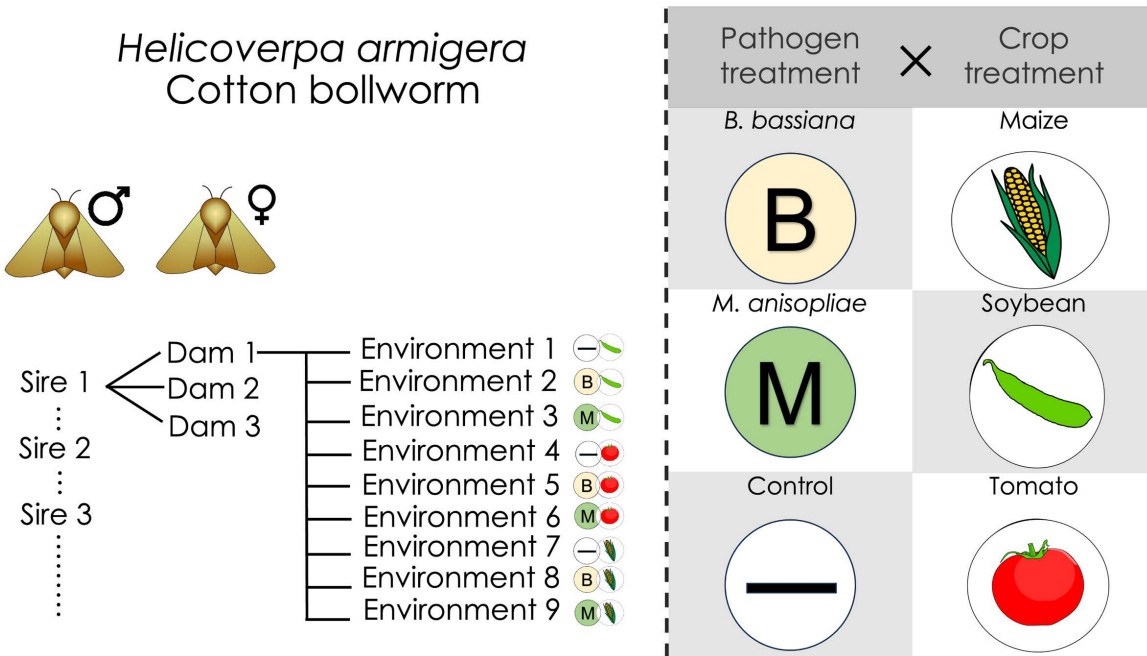

**Fig 5. Schematic representation of the experimental design.** Second instar larvae from each female were randomly assigned to one of 9 treatments. Insects were exposed to one of three different infection treatments and were reared on one of three different plants. Larval survival was recorded daily thereafter.

If unsuccessful, moths were re-paired and observed as before. Mating was deemed successful if the moths remained attached for more than 20 min and then separated successfully. Females that had successfully mated were transferred to ventilated plastic boxes (55 mm (l) × 55 mm (w) × 60 mm (h)) for oviposition and fed on cotton wool soaked with vitamin solution [71].

*H. armigera* egg masses were collected daily from each female, split approximately evenly into three groups, and placed in ventilated plastic boxes (7 cm × 7 cm × 7 cm) containing either fresh maize, soya or tomato leaves until hatching. Fresh leaf material was added daily. Early second instar *H. armigera* larvae from each plant treatment were randomly assigned to one of three infection treatments with sunflower oil spore suspensions: *B. bassiana* IBCB1363, *M. anisopliae* IBCB 425 and a pathogen-free control. Larvae were placed individually in Petri dishes (4.5 cm diameter) and 0.5 µl spore suspension (or blank oil) was pipetted onto the larval cuticle. Larvae were left for 2 hrs, then transferred to individual plastic vials (23 ml, Sarstedt), sealed with breathable cellulose acetate flugs and provided with fresh leaves of the plant treatment they had previously fed on. Larvae were held at 25°C, 75% R.H. and 14:10 hr L:D). Mortality of larvae was first recorded 24 hrs after treatment and then daily thereafter until death or pupation; larvae were transferred using a sterilized fine brush onto a fresh leaf diet when required. This experiment was conducted over two blocks, with identical experimental design for each; block 1 used 18 sires (32 dams), and block 2 had 19 sires (26 dams).

## Data analysis

From each dam, we attempted to rear 90 offspring, though in some cases we obtained fewer than the necessary number of eggs. Consequently, our preliminary offspring count was 4344 larvae, of which 4314 survived to day 3 when the infection/control treatment was applied. To avoid error in quantifying mortality variation that was not due to pathogens, we excluded a small proportion of larvae that died shortly after inoculation in a way that was unlikely to reflect pathogen

infection. For example, in some cases (7.8%, N = 339) death occurred within two days of treatment, a time when fungal infection is unlikely to have proceeded to the lethal stage. In other cases (3.7%, N = 162), the larvae never moved after being treated and so may have died immediately following oil application. To prevent these instances from obscuring patterns that were due to the treatments, we removed both categories before analysis. A small number of larvae (N = 2) also escaped before treatment. This left us with 3811 larvae included in the final dataset.

We performed all statistical analyses using R4.3.2 [73]. We computed mortality rates at daily intervals from day 5 after infection through day 14 (a range that should capture most of the relevant pathogen-induced mortality) and we chose the day on which distinctions between control treatments and pathogen treatments were highest (day 14), to most closely capture differences in genetic variation for pathogen susceptibility.

To describe patterns of mortality in pathogen and plant diet treatments, we built a generalised linear model with logit link implemented in lme4 [74] in which both pathogen and plant-diet treatments as well as their interaction were predictors of the binomial survival proportion. We fitted random effects for both sire and dam identity to account for non-independence of larvae due to family membership and to control for maternal effects. We used parametric bootstrapping [75] to test the significance of the plant:pathogen treatment interaction when comparing nested models.

To compute quantitative genetic parameters, we fit generalised models using the brms package [76]. Bayesian analyses are uniquely suited for fitting complex models with many parameters and quantifying uncertainty in these estimates, especially when values of some parameters hinge on values of others, as is true when estimating genetic covariances. We fitted the combination of plant diet and pathogen treatment as a 9-level fixed factor and fitted insect sires and dams as random effects. We further allowed the effect of sire to vary by treatment and asked the model to estimate correlations across treatments to reconstruct the G-matrix. We did not similarly allow dam effects to vary by treatment, because effects related to maternal condition (e.g., through egg provisioning) should uniformly improve offspring survival regardless of the specific treatment. Note that in our experiments dams were never exposed to pathogens and were fed an artificial diet, so there is no potential for interesting trans-generational maternal advantages due to phenotype matching. To ensure sufficient warmup and chain mixing, we ran the models for 32,000 iterations (half of which were used as warmup iterations) and adjusted the no-U-turn sampler (NUTS) by setting adapt_delta to 0.96 to avoid divergent transitions. Our models produced well-mixed chains and unimodal posterior distributions. The estimates were robust to minor changes in prior specifications and were not influenced noticeably by alternate model structures (e.g., fitting only two treatments to produce a single genetic correlation estimate instead of estimating all 36 estimates simultaneously from a single model).

We used samples from the posterior to compute heritabilities within each combination of plant diet and pathogen treatment. For each sample, the environment-specific heritability was calculated as four times the sire variance in that environment (to reflect the fact that sires share a quarter of genes with their offspring in a half-sibling design) divided by the sum of four times the sire variance, the maternal variance, and the residual variance (fixed at $\pi^2/3$ because this is a logistic model) in that posterior draw [77]. We also used posterior samples to compute genetic correlations; these are extracted for each pair of environments, and since there are 9 environments there are 36 pairwise combinations. We analysed patterns for the genetic correlations with respect to change in plant diet, pathogen treatment, and the combination of the two. To highlight the Bayesian nature of these analyses we report 89% highest posterior density intervals (89% HPDI, in contrast to 95% confidence intervals) when comparing different environmental contrasts, but we note that the complete posterior distribution is the best representation of *a posteriori* evidence [78]. For this reason, to facilitate an appreciation of the total evidence we illustrate posterior densities using ridgeplots [79].

## Supporting information

**S1 Table. Impact of crop diet and fungal pathogen on larval survival: GLMM estimates.**
(DOCX)

**S1 Fig. Relative ability of fungal isolates to kill *Helicoverpa* larvae depended on crop leaf diet and experimental block.**
(TIF)

**S2 Fig. Infection treatment alters the relative fitness of different half-sibling families.**
(TIF)

**S3 Fig. Diet treatment alters the relative fitness of different half-sibling families.**
(TIF)

**S4 Fig. Pathogen and diet treatment alter the relative fitness of different half-sibling families.**
(TIF)

**S5 Fig. Rank order of fitness for different half-sibling families is disrupted by changes in diet and infection treatment.**
(TIF)

## Acknowledgments

We are grateful to James Weir for technical support and management of controlled environment facilities.

## Author contributions

**Conceptualization:** Matthew C. Tinsley, Ricardo A. Polanczyk, Luc F. Bussière.

**Data curation:** Rosie M Mangan.

**Formal analysis:** Rosie M Mangan, Matthew C. Tinsley, Luc F. Bussière.

**Funding acquisition:** Rosie M Mangan, Matthew C. Tinsley, Ricardo A. Polanczyk, Luc F. Bussière.

**Investigation:** Rosie M Mangan, Ester Ferrari.

**Methodology:** Rosie M Mangan, Matthew C. Tinsley, Ester Ferrari, Luc F. Bussière.

**Project administration:** Rosie M Mangan, Luc F. Bussière.

**Resources:** Rosie M Mangan.

**Supervision:** Matthew C. Tinsley, Luc F. Bussière.

**Visualization:** Rosie M Mangan, Matthew C. Tinsley, Luc F. Bussière.

**Writing – original draft:** Rosie M Mangan, Matthew C. Tinsley, Luc F. Bussière.

**Writing – review & editing:** Rosie M Mangan, Matthew C. Tinsley, Ricardo A. Polanczyk, Luc F. Bussière.

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
