## [Decision Letter · Decision Letter 0]

6 Mar 2025

PPATHOGENS-D-25-00219

Exploiting pathogen defence trade-offs to manage risks of crop pest evolution against biocontrol agents

PLOS Pathogens

Dear Dr. Mangan,

Thank you for submitting your manuscript to PLOS Pathogens. After careful consideration, we feel that it has merit but does not fully meet PLOS Pathogens's publication criteria as it currently stands. Therefore, we invite you to submit a revised version of the manuscript that addresses the points raised during the review process.

Please submit your revised manuscript within 30 days May 05 2025 11:59PM. If you will need more time than this to complete your revisions, please reply to this message or contact the journal office at plospathogens@plos.org. Please include the following items when submitting your revised manuscript:

We look forward to receiving your revised manuscript.

Kind regards,

Pedro F Vale

Academic Editor

PLOS Pathogens

Debra Bessen

Section Editor

PLOS Pathogens

Sumita Bhaduri-McIntosh

Editor-in-Chief

PLOS Pathogens

orcid.org/0000-0003-2946-9497

Michael Malim

Editor-in-Chief

PLOS Pathogens

orcid.org/0000-0002-7699-2064

**Additional Editor Comments (if provided):**

Dear Dr Mangan,

Thank you for submitting your work to PLOS Pathogens which has now been assessed by three expert reviewers. All reviewers were very positive about this work, citing the importance of the question, the ambition and scale of the work and the rigorous handling of the data and write-up. I was also impressed by the scope of the work. The reviewers did however highlight a few issues that require some work to improve the manuscript.

One point was the sometimes confusing nature of the terminology, given that resistance can refer to the pathogen or the pest. One reviewer suggests this may be addressed by using 'susceptibility' as an alternative, to avoid issues with the use of resistance in other contexts. Overall it would be advisable to revise the terminology used to avoid confusion, especially given the tri-trophic nature of the system.

Another point are some potential blindspots in covering previous work on pest biocontrol and pathogen resistance. Please make sure to incorporate this work into the current manuscript where appropriate.

Overall, these issues would seem straightforward to address in a revised version.

**Journal Requirements:**

**Reviewers' Comments:**

Reviewer's Responses to Questions

**Part I - Summary**

Reviewer #1: Please see Part III.

Reviewer #2: Mangan et al. investigates timely and important topic how resistance to biocontrol agents evolves in multi host cropping. They approached topic from both theoretical and applied perspective and their experimental design was considerably large (3800 larvae) with three host species and two biocontrol agents. The experimental design seems appropriate and analysis rigorous. They find heritable variation in larvae ability to survive biopesticides. The main novelty in the manuscript is that the authors test whether host species and biopesticide identity can alter that variation. They find that there is only lesser extent of variation caused by biopesticide identity, but host species alters selection. They discuss theoretical and crop protection implications for their results. The manuscript is generally well written, and I do not have any major criticism.

Reviewer #3: Exploiting pathogen defence trade-offs to manage risks of crop pest evolution against

biocontrol agents

I really enjoyed reading this MS- it was exceptionally well written and addresses an interesting and important aspect of resistance to biocontrol agents and the genetics of pathogen susceptibility.

Strengths

The major novel aspect of this study was the size and ambition of the quantitative genetic analysis of pathogen susceptibility. Assessing even simple life history data across three host plants and two pathogens is a major undertaking and the authors should be congratulated on achieving this.

Weakness

A few issues stood out. First was the authors' very casual use of the term 'resistance'. This can be a loaded term and to many in pest management it implies that a control strategy has been launched and then subsequently has failed due to some recent genetic change.

Without doubt the authors have data on variation in pathogen susceptibility but it is a stretch to describe this as 'resistance'. I would prefer it if the authors refer to their data study as variation in susceptibility and I suspect many in the field would agree with me.

This distinction is important genetically. For example, I suspect that it's no coincidence that pathogen mortality varies from 0-100%- the authors have perhaps chosen a dose that gives the most information.

This is important as in biocontrol doses are generally chosen to give as little variability as possible. Importantly, as doses increase we would expect fewer and fewer loci to contribute to variation in susceptibility - this is an important (and perhaps unavoidable) limitation of the study and should be discussed.

The authors mention polygenic resistance for pathogens- but I am not sure how well this is borne out in the field - Bacillus thuringiensis resitance is typically monogenic (references too numerous to mention); resistance to Cydia pomonella granulosis virus (the other classic example of field resistance to a pathogen biocontrol agent) is also apparently monogenic https://doi.org/10.3390/v9090250. Here, weak resistance in laboratory selection is often misleading but even so earlier virus work found monogenic resistance https://doi.org/10.1016/0022-2011(82)90013-1. Perhaps fungi are special here but this should have a more specific discussion.

The second major issue is that I feel the authors have neglected to cite and discuss a significant body of work on pathogen resistance - most of this relates to work on resistance to Bacillus thuringiensis, but there is also a wider and perhaps pertinent literature on how host plants affect susceptibility to virus (NPVs, GVs) that the authors might want to consider.

For instance, one suggestion the authors make (line 360-365)- applying different pathogens to different refugia as a means of resistance management - this has actually been tested experimentally (doi: 10.1111/j.1365-2664.2007.01285.x) - see also work on Bt resistance and susceptibility to nematodes (doi: 10.1111/j.1365-2664.2008.01457.x). These results challenge the generality of the concluding paragraph here ie the expection for negative trade-offs between pathogens. Importantly, previous studies have used very different pathogens/parasites and not two Entomophoralean fungi with a similar mode of action.

Note also research on fitness costs to Bt In terms of NCCRs between resistance and performance on different plants has quite a substantial literature. For example, within crop variation in Brassicas is such that it is possible to find varieties that are nearly lethal to Bt resistant genotypes doi:10.1038/hdy.2010.65- that study also showed that lower quality plants increased fitness costs of resistance across multiple insect genotypes. see also Janmaat AF, Myers JH (2005). The cost of resistance to Bacillus

thuringiensis varies with the host plant of Trichoplusia ni. Proc R Soc Lond Ser B-Biol Sci 272: 1031–1038. Those two aforementioned studies found that resistant insects tended to do worse on plants of lower quality- something the authors seem not to have repeated- that is an important topic for the discussion in itself.

Minor comments

NCCR- Agree that its rare but also see classic example from Blackfly that was incorporated in resistance management scheme : Kurtak, D., Meyer, R., Orcran, M., & Tele, B. (1987). Management of insecticide resistance in the control of Simulium damnosum complex

Medical and Veterinary Entomology, 1, 137–146.

**Part II – Major Issues: Key Experiments Required for Acceptance**

Reviewer #1: N/A

Reviewer #2: (No Response)

Reviewer #3: na

**Part III – Minor Issues: Editorial and Data Presentation Modifications**

Reviewer #1: This excellent paper reports rigorous experiments showing genetic tradeoffs across

crop plants affecting the ability of a major pest to resist two fungal pathogens. The

results have important implications for fundamental understanding of evolution of

pathogen-host interactions and enhancing sustainability of crop protection.

I have only minor suggestions for improvement:

Please address the classic tactic of using non-host plants in rotations or spatial mosaics

with host plants to suppress pests, including pros and cons versus using only host

plants of pests as evaluated here.

This paper is about pathogens. Accordingly, for clarity and simplicity, please use the

term pathogen throughout rather than alternating between parasite and pathogen. If

there is a specific reason to consider parasites that are not pathogens in some places,

explain that explicitly.

Several terms in the title are ambiguous in the absence of context. Suggested revised

title to improve clarity: Delaying resistance to biopesticides by exploiting trade-offs

across crops in pest susceptibility to pathogens.

Be clear in the introduction, abstract, and elsewhere that the paper addresses variation

in the pest’s susceptibility (or resistance) to fungal pathogens. This isn’t obvious from

the term “pathogen defence.”

L39 & 41 Revise to “genetic variation for resistance to pathogens.”

L53. Revise to “such as biopesticides containing live pathogens.”

Cite a more apt reference: Microbial biopesticides in agroecosystems. Agronomy 2018,

8, 235; doi:10.3390/agronomy8110235

L75 “Defence against microbes is typically more genetically complex than for synthetic

insecticides or genetically modified crops[8, 21, 22].” Cite one or more current, rigorous

references to support this claim or omit it. This claim does not seem essential here and

actually detracts from the paper’s main thrust. I could not find credible, rigorous

evidence in the references cited: 8, 21 or 22 (or 4). One systematic review to consider:

X.-P. Lu, L. Xu, J.-J. Wang, Mode of inheritance for pesticide resistance, importance

and prevalence: A review. Pestic. Biochem. Physiol. 202, 105964 (2024). Contrary to

the claim above, this review reports ca. 60% of the 187 cases of lab- and field-selected

pesticide resistance (mostly to synthetic insecticides) reviewed were “polygenic.” Also, 3

of the 4 cases of resistance to microbes (Bt or Bacillus sphaericus) reviewed were

“monogenic.” I think negative cross-resistance has not been widely applied because it

is rare. No need to invoke genetic complexity to explain this.

As noted in the Discussion, the references cited for genetic complexity in the

introduction do not support the claim of defense trade-offs. So, if there is evidence that

resistance to pathogens is typically complex, the inference that such complexity

promotes tradeoffs is not supported by the references cited. So, either provide clear

support for the claim that genetic complexity promotes trade-offs or omit the claim.

The evidence cited from D. melanogaster shows a positive correlation in defense across

bacteria and fungi, so why is it surprising that a negative correlation was not seen

between two fungal pathogens? Before publication of the D. melanogaster papers in

2017 and 2023, it might have been reasonable to expect tradeoffs between fungal

pathogens. But this is no longer true and the paper should recognize this evidence and

reflect the appropriate change in perspective throughout.

L413. To avoid overstatement, change “answered” to “addressed.”

L433. “However, our data call into question the prevailing expectation that trade-offs in

host resistance to different pathogens have a prime role in maintaining genetic variation

for pathogen defence traits.” The previous results from D. melanogaster also refute this

expectation and should be cited here.

Please consider and cite related work on Bt, which is the most studied and widely used

biopesticide (used for nearly a century, including decades before transgenic crops). For

example, see: Ecol. Ent. 2006, https://doi.org/10.1111/j.0307-6946.2006.00768.x

Annu. Rev. Entomol. 2009, https://doi.org/10.1146/annurev.ento.54.110807.090518

and J. Econ. Entomol. 2024, https://doi.org/10.1093/jee/toae077.

Reviewer #2: (No Response)

Reviewer #3: The figure legends were not easy to match to figures as they occured at various intervals in text - this did not help reviewing.

PLOS authors have the option to publish the peer review history of their article (what does this mean? ). If published, this will include your full peer review and any attached files.

**Do you want your identity to be public for this peer review?** For information about this choice, including consent withdrawal, please see our Privacy Policy .

Reviewer #1: No

Reviewer #2: **Yes: ** Hanna Susi

Reviewer #3: No

**Figure resubmission:**
---

## [Editor Report · Decision Letter 1]

19 Apr 2025

Dear Dr Mangan,

We are pleased to inform you that your manuscript 'Crop diversity induces trade-offs in microbial biopesticide susceptibility that could delay pest resistance evolution' has been provisionally accepted for publication in PLOS Pathogens.

Best regards,

Pedro F Vale

Academic Editor

PLOS Pathogens

Debra Bessen

Section Editor

PLOS Pathogens

Sumita Bhaduri-McIntosh

Editor-in-Chief

PLOS Pathogens

orcid.org/0000-0003-2946-9497

Michael Malim

Editor-in-Chief

PLOS Pathogens

orcid.org/0000-0002-7699-2064

Dear Dr Mangan,

Thank you for submitting your revised manuscript. I was pleased to see that all the reviewer comments were addressed with great care and in such great detail.

The question of the genetic complexity underlying resistance is certainly a point that generated much discussion, and I appreciated the lengthy and detailed explanation for your rationale. On this point, I think the changes made directly in the manuscript do a good job of justifying your position, especially with the addition of new cited references. I think I agree with the statement that this topic would possibly warrant a paper on it's own - perhaps something to consider for another venue.

I also appreciate that you identified and corrected the error in the calculation of the additive variance for the logistic model. This error is actually quite commonly found in published work, and while estimates tend to be similar (as in this case), it is important that this is done correctly.

Overall, I am very happy with the extent of the revisions, and I congratulate you for this excellent contribution.

Sincerely,

Pedro F Vale

Academic Editor

PLOS Pathogens
---

## [Editor Report · Acceptance letter]

Dear Dr Mangan,

We are delighted to inform you that your manuscript, "Crop diversity induces trade-offs in microbial biopesticide susceptibility that could delay pest resistance evolution," has been formally accepted for publication in PLOS Pathogens.

Best regards,

Sumita Bhaduri-McIntosh

Editor-in-Chief

PLOS Pathogens

orcid.org/0000-0003-2946-9497

Michael Malim

Editor-in-Chief

PLOS Pathogens

orcid.org/0000-0002-7699-2064